# Bioactive-Based Cosmeceuticals: An Update on Emerging Trends

**DOI:** 10.3390/molecules27030828

**Published:** 2022-01-27

**Authors:** Anju Goyal, Aditya Sharma, Jasanpreet Kaur, Sapna Kumari, Madhukar Garg, Rakesh K. Sindhu, Md. Habibur Rahman, Muhammad Furqan Akhtar, Priti Tagde, Agnieszka Najda, Barbara Banach-Albińska, Katarzyna Masternak, Ibtesam S. Alanazi, Hanan R. H. Mohamed, Attalla F. El-kott, Muddaser Shah, Mousa O. Germoush, Hamdan S. Al-malky, Salman H. Abukhuwayjah, Ahmed E. Altyar, Simona G. Bungau, Mohamed M. Abdel-Daim

**Affiliations:** 1Chitkara College of Pharmacy, Chitkara University, Rajpura 140401, India; anju.goyal@chitkara.edu.in (A.G.); adisharma96@gmail.com (A.S.); jashankaur159@gmail.com (J.K.); ms.sapnakumari92@gmail.com (S.K.); madhukar.garg@chitkara.edu.in (M.G.); 2Department of Global Medical Science, Yonsei University Wonju College of Medicine, Yonsei University, Wonju 26426, Korea; 3Department of Pharmacy, Southeast University, Dhaka 1213, Bangladesh; 4Riphah Institute of Pharmaceutical Sciences, Lahore Campus, Riphah International University, Lahore 54000, Pakistan; mfurqan.akhtar@riphah.edu.pk; 5Amity Institute of Pharmacy, Amity University, Noida 201303, India; tagde_priti@rediffmail.com; 6Department of Vegetable and Herbal Crops, University of Life Sciences in Lublin, 20-280 Lublin, Poland; agnieszka.najda@up.lublin.pl; 7Department of Zoology, Animal Ecology and Wildlife Management, University of Life Sciences in Lublin, 20-950 Lublin, Poland; barbara.banach@up.lublin.pl; 8Institute of Plant Genetics, Breeding and Biotechnology, University of Life Sciences in Lublin, 20-950 Lublin, Poland; katarzyna.masternak@up.lublin.pl; 9Department of Biology, Faculty of Sciences, University of Hafr Al Batin, Hafr Al Batin 39923, Saudi Arabia; esalanazy@uhb.edu.sa; 10Zoology Department, Faculty of Science, Cairo University, Giza 12613, Egypt; hananeeyra@cu.edu.eg; 11Biology Department, Faculty of Science, King Khalid University, Abha 61421, Saudi Arabia; elkottaf@kku.edu.sa; 12Zoology Department, College of Science, Damanhour University, Damanhour 22511, Egypt; 13Department of Botany, Abdul Wali Khan University Mardan, Mardan 23200, Pakistan; muddasershah@awkum.edu.pk; 14Biology Department, College of Science, Jouf University, Sakaka 42421, Saudi Arabia; mogermoush@ju.edu.sa; 15Regional Drug Information Center, Ministry of Health, Jeddah 21442, Saudi Arabia; hamdan27@hotmail.com; 16Pharmacy Program, Batterjee Medical College, Jeddah 21442, Saudi Arabia; salman@bmc.edu.sa; 17Department of Pharmacy Practice, Faculty of Pharmacy, King Abdulaziz University, Jeddah 21589, Saudi Arabia; aealtyar@kau.edu.sa; 18Department of Pharmacy, Faculty of Medicine and Pharmacy, University of Oradea, 410304 Oradea, Romania; simonabungau@gmail.com; 19Department of Pharmaceutical Sciences, Batterjee Medical College, Jeddah 21442, Saudi Arabia; 20Pharmacology Department, Faculty of Veterinary Medicine, Suez Canal University, Ismailia 41522, Egypt

**Keywords:** cosmetic-containing herbals, bioactive ingredients, cosmetics

## Abstract

Cosmetic-containing herbals are a cosmetic that has or is claimed to have medicinal properties, with bioactive ingredients purported to have medical benefits. There are no legal requirements to prove that these products live up to their claims. The name is a combination of “cosmetics” and “pharmaceuticals”. “Nutricosmetics” are related dietary supplements or food or beverage products with additives that are marketed as having medical benefits that affect appearance. Cosmetic-containing herbals are topical cosmetic–pharmaceutical hybrids intended to enhance the health and beauty of the skin. Cosmetic-containing herbals improve appearance by delivering essential nutrients to the skin. Several herbal products, such as cosmetic-containing herbals, are available. The present review highlights the use of natural products in cosmetic-containing herbals, as natural products have many curative effects as well as healing effects on skin and hair growth with minimal to no side effects. A brief description is given on such plants, their used parts, active ingredients, and the therapeutic properties associated with them. Mainly, the utilization of phytoconstituents as cosmetic-containing herbals in the care of skin and hair, such as dryness of skin, acne, eczema, inflammation of the skin, aging, hair growth, and dandruff, along with natural ingredients, such as for hair colorant, are explained in detail in the present review.

## 1. Introduction

The cosmetic-containing herbals market is rapidly growing globally among the naturals plend or industry [1]. The word cosmetic is derived from the Greek word “kosm tikos”, which means power, arrangement, and skill in decoration. The origin of cosmetics forms a constant narrative throughout the history of man as they developed. These are topical corrective pharmaceutical combos proposed to complement beauty by way of the utilization of their components that have suitable characteristics required for the care of skin and hair. According to The Drugs and Cosmetics Act, 1940, cosmetics are defined as articles meant to be rubbed, poured, sprinkled, or sprayed on, introduced in to, or in any other case applied to the human frame for cleaning, beautifying, promoting attractiveness, or altering appearance. Herbal cosmetics are products that are formulated by the use of a cosmetic substance that acts as a base in which natural ingredients that have health benefits are added [1,2].

Cosmetic-containing herbals are a combination of cosmetics and pharmaceuticals, incorporated with biologically active phytochemicals that are responsible for the therapeutic effect of the formulation. They are only applied topically such as creams, ointments, and lotions. Currently, skin diseases are common among all age groups. They are mainly associated with exposure to microbes, biological toxins, and chemical agents. Sometimes, malnutrition also has a very extensive role in the development of skin diseases [3]. The beauty of skin and hair is mainly based on health, habits, maintenance, and climatic conditions. The science of Ayurveda classified extensive plants and floras for their effects in the beautification and protection of skin from external hazards. This review mainly emphasizes the plants and phytoconstituents that possess skin protection properties.

Advantages of herbal cosmetics [2,3]:Decreases the threat of an unfavorably prone response and does not have negative symptoms;Easily incorporates into the hair and skin;Has better patient tolerance and acceptance;Acts as a renewable source of medication;Extensive availabilities, especially in developing countries such as India;Effective in small portions when contrasted with manufactured beauty care merchandise;Extracts of vegetation decreases the bulk belongings of cosmetics and provides suitable pharmacological results;Effectively accessible and found in a vast assortments and amounts;Less expensive.

Disadvantages of herbal cosmetics [2,3]:The interest in herbal tablets is growing comparatively slower than allopathic medicines;Overlaying the taste and odor is difficult;The availability of herbal tablets is limited;Manufacturing procedures are tedious and convoluted;No pharmacopeia characterizes a particular technique or fixings to beutilized by any of the natural beautifying dealers.

## 2. Regulation and Licensing of a Cosmetic Containing Herbals

A cosmeceuticals registration process should be simpler than that of pharmaceuticals such as tablets. Of course, according to good clinical practices (GCPs), clinical studies with adequate results are required to demonstrate the cosmeceutical’s intended activity for the treatment of a specific minor skin disorder or “condition”, and there must be assurance that safety requirements are met and that no expected side effects are present [4]. In the United States, a subclass of pharmaceuticals (cosmeceuticals) is registered in the same way as over-the-counter products. The high-profile court battle over the regulatory labeling of topical minoxidil for male pattern baldness resulted in the argument that whether a product is a cosmetic or a medication is determined by its medicinal action, not the condition it is meant to treat (normal vs. ailing skin) [5].

## 3. Preparation of Herbal Extract

Botanicals must receive a considerable level of chemical processing before being included in a cosmeceutical, and this process has a significant influence on the botanical’s biological value. The source of the plant material is the most significant aspect to be considered for the biological use of an herb, with it being incorporated into a cosmeceutical-containing herbal product, since each plant piece may contain a variety of chemical compounds and molecules. In addition, developing conditions, such as the structure of the soil, water availability, trade in weather conditions, plant stress, and gathering conditions, including time from harvesting time, transit time of harvested fabric, plant fabric care for the duration of transportation, stockpiling conditions preceding assembling, and preparation of the herb extract and, ultimately, the finished product, are considered as other significant requirements. Moreover, other factors or natural elements that may drastically affect solubility, stability, biologic availability, pharmacokinetics, pharmacologic pastime, and toxicity must be considered while selecting an herb [6].

Crushing, grinding, comminuting, boiling, distilling, pressing, drying, or exposing them to solvents are methods used to obtain galenic extracts from leaves, roots, end products, berries, stems, twigs, barks, and plants. In most cases, the plant fabric is heated or arranged in order to extract essential oils or other distillates that may be easily included into a cosmetic product. However, certain physiologically active molecules may be terminated or altered as a result of this processing. Oil, wax, juice, tincture, decoction, tea, infusion, and/or powders are the end products, which are subsequently formed into topical applications such as solutions, gels, lotions, ointments, and pastes. Some of these preparations are used in the same way as fomentation, compresses, and poultices. The medicinal value of any plant is influenced by the amount of attention it receives. The restorative botanicals are usually supplied in small, sub-remedial amounts in cosmetic-containing herbals. Few herbs are required in a low awareness situation to obtain the desired effect because their potency is exceptionally high. The stratum corneum functions as a permeability barrier, allowing natural items to be tested for performance. Botanicals include many energetic additives with varying solubilities, polarities, and healing foci, making transport over the mucocutaneous surface difficult [6,7].

The additives used in cosmetics/cosmetic-containing herbals are shown in Table 1.

## 4. Skin Cosmetic-Containing Herbals

Cosmetic-containing herbals are cosmetic items that have healing or medication-like benefits that can have an impact on the biological workings of the pores and skin according to their substances. They are whole some skin items that go beyond shading and improving pores and skin; they improve the running/surface of the skin by way of enabling collagen advancement by battling the volatile effects of unfastened radicals, hence, maintaining keratin’s shape in notable circumstance and making the skin healthier as shown in Table 2 and Table 3 [15,16].

## 5. Fixed Oils

### 5.1. Coconut Oil

Biological Source: Coconut oil is acquired from the dried, dense parts of the endosperm of coconut. *Cocos nucifera* belongs to the own family Palmae. Its chemical constituents include oil, which accounts for approximately 95% of the saturated fatty acid with 8–10 carbonatoms. This indicates the presence of caprylic acid, 2% picric acid, 50–80% lauric acid, and 27% myristic acid (1%) (Figure 1). It is used as a high-quality pore and skin moisturizer and softener [20,21].

### 5.2. Almond Oil

Biological Source: Almond oil is a fixed oil procured with the aid of articulation from the seeds of *Prunus amygdalus* (Rosaceae) var. *dulcis* (candyalmond) or *P. Amygdalus* var. Amara (bitter almond) [22]. Its chemical constituents are a blend of glycerides of oleic, linoleic, palmitic, myristic, palmitoleic, oleic, margaric, stearic, linolenic, linoleic, arachidonic, gadoleic, behenic, and erucic acids (Figure 2). It is used as an emollient as well as in cosmetic arrangements [22].

### 5.3. Olive Oil

Biological Source: Olive oil is a fixed oil procured by articulation of the ripeculmination of *Olea europaea* or *Indian olive* (Olea ferruginea), belonging to the family Oleaceae. Its chemical constituents feature substantial components of tocopherol, monostearate, triolein, tripalmitin, trilinolein, tristearate, triarachidin, squalene, and β-sitosterol (Figure 3).It is used as an emollient or skin conditioner in cosmetic-like lotions [23].

### 5.4. Sesame Oil

Biological Source: Sesame oil is obtained by means of extracting and refining the oil from the seeds of *Sesamum indicum*, which is from the family Pedaliaceae. Its chemical constituents comprises blend of glycerides of oleic, linoleic, palmitic, stearic, arachidic, hexadecenoic, lignoceric, and myristic acids (Figure 4). In addition, it ac commodities lignan, sesamin, the associated sesamolin, and vitamins A and E [24]. It is used as an emollient. It is also an extract that is rich in lignans that have antioxidant and radical scavenger uses and is used in the beauty industry [24].

### 5.5. Castor Oil

Biological Source: Castor oil is a fixed oil received by the blood less expression of the seeds of *Ricinus communis*, belonging to the family Euphorbiaceae. Its chemical constituents include glycerides of ricinoleic acid (Figure 5), iso-ricinoleic, stearic, and dihydroxy stearic acids. It also consists of diet F [25]. It is used for its anti-microbial and anti-inflammatory properties, making it helpful in decreasing pimples. Castor oil is rich in fatty acids, thus enhancing smoothness and softening while being able to implement facialpores and skin. It promotes an increase in healthy pores and skin tissue through moisturizing, hydrating, and cleansing [25].

## 6. Waxes

### 6.1. Beeswax

Biological Source: Refined bees wax is received from honeycombs of honeybee hives, particularly *Apis mellifera* and, a different pecies of Apis inside the circle of its relatives Apidae. Its chemical constituents include myricin, loose cerotic acid, and myricyl alcohol. It is used to maintain moisture and acts as a skin softener. It has anti-inflammatory and antibacterial properties. It also works as an antioxidant [23,26].

### 6.2. Carnauba Wax

Biological Source: Carnauba wax exudates from the apertures of the leaves of the Brazilian wax-palm trees *Copernicia prunifera* and *C. Cerifera* from the family Palmae. Its chemical constituents include esters of hydroxylated fatty acids, i.e., carnaubic and cerotic acid (Figure 6) and melissylcerotate. It is used for the preparation of cosmetic products such as lipsticks and lip balms [27].

## 7. Gums

### 7.1. Tragacanth

Biological Source: Tragacanth is air-dried gummy exudate that flows naturally oris received by incision from the stems and branches of *Astragalus gummifer* belonging to the circle of relatives Leguminosae. Its chemical constituents comprise important fractions—first being water soluble, termed as tragacanthin, and the second being water insoluble, termed as bassorin. It is used as an emollient in cosmetics, and as a thickening, postponing, and emulsifying agent [28].

### 7.2. Agar

Biological Source: Agar is a dried coagulated substance obtained through extraction with water from *Gelidiumamansi* or an exceptional form of pink algae such as Gracilaria and Pterocladia, which belong to families Gelidiaceae (Gelidium and Pterocladia) and Gracilariaceae (Gracilaria). Its chemical constituents incorporate exclusive poly saccharides referred to as agarose and agaropectin. Agarose is liable forits gelproperty and agar pectin for its viscosity. It is used as a gelling agent in cosmetics [29].

## 8. Perfumes

Perfume can be obtained from various sources as shown in Table 4.

### 8.1. Rose Oil

Biological Source: Rose oil is an essential oil acquired from the petals of exclusiverose species, *Rosa damascene* and *Rosa centifolia*, which belong to the family Rosacea. Its chemical constituents include β-damascenone, β-damascone, β-ionone (Figure 7), and rose oxide, which account for its scent [30]. It is used in perfumeryin addition to cosmetics.

### 8.2. Lavender Oil

Biological Source: Lavender oil is vital oil acquired by distillation from the flower spikes of certain species of lavender, viz, *Lavandula limon officinalis* which belongs to the circle of its relatives Lamiaceae. Its chemical constituents include alpha-pinene, limonene, 1,8-cineole, cis-ocimene trans-ocimene,3-octanone, camphor, linalool, caryophyllene, and lavandulyl acetate (Figure 8) [31]. It has antiseptic and anti-inflammatory properties, and it is used in perfumes, cosmetics, and topical software, as well as in the treatment of pores and skin for burns to help ease the pain and to treat skin infections [31].

### 8.3. Immortelle Oil

Biological Source: Immortelle oil is vital oil that is light yellow to pink in color. It is a slick fluid with a stable nectar-like smell received from the plant *Helichrysum italicum*. Its chemical constituents contain neryl acetate, curcumene, neryl propionate, ar-curcumene, geraniol, geranyl acetate, nerol, (Figure 9) etc. It is used as aperfumein cosmetics and perfumery. It prevents any type of pore or skin infection or contamination caused by bacteria or fungi due to the fact of its antimicrobial and antifungal properties [30].

### 8.4. German Chamomile Oil

Biological Source: German chamomile oil is a critical oil obtained by way ofthe steam distillation of the flower and flower buds of *Matricaria chamomilla* [32]. Its chemical constituents consist of active ingredients that include β-farnesene, farnesol, chamazulene (Figure 10), α-bisabolol oxides A and B that have anti-inflammatory, antiphlogistic, and antiseptic properties [7,32]. It is used in skin lotions, skin oils, decorative cosmetics, and shampoos [7].

### 8.5. Neroli Oil

Biological Source: Neroli oilis a vitaloil extracted from the flower of *Citrus aurantium* var. Amara, additionally referred to as bitter orange. Its chemical constituents include linalool, beta-pinene, alpha-terpineol, limonene, nerol, sabinene, nerolidol, linalyl acetate, and alpha-pinene (Figure 11). It is used in cosmetics, and it is normally used to refresh either touchy or oily tired skin.

### 8.6. Rosemary Essential Oil

Biological Source: Rosemary essential oil is produced through steam distillation of the flowering hints of the plant *Rosamarinus officinalis* that belongs to the family Lamiaceae [33]. Its chemical constituents include the main chemicals of eucalyptol and alpha-pinene. Camphor, bornyl acetate, beta-pinene, limonene, and borneol (Figure 12) alsomake up the oil. It is found in cosmetic merchandise that includes slvender water. This oil is widely used for hair care, because it nourishes the hair, promotes hair growth, and helps again stdandruff. It is recommended in hair-loss remedies, as it is believed to have similar characteristics to minoxidil [33].

**Table 4 molecules-27-00828-t004:** Various sources of perfume and its examples.

Perfume Sources	Examples	References
Natural animal source	Musk, Civet, Ambergris, Castoreum, etc.	[34]
Natural plant source	Rose, Jasmine, Lavender, Lemon, etc.	[34]
Aroma chemical	Eugenol, Farnesal, Rose oxide, Citral, Limonene	[35]
Floral base	Rose and Jasmine	[36]
Woody base	Citrus, Oriental, Fruity, etc.	[36]

### 8.7. Tea Tree Oil

Biological Source: Tea tree oil is the essential oil acquired by distillation from the leaves and terminal branch lets of *Melaleuca alternifolia* [7].

Its chemical constituents include alpha-pinene, beta-pinene, sabinene, myrcene, alpha-phellandrene, alpha-terpinene, limonene, 1,8-cineole, p-cymene, linalool, terpinen-four-ol, alpha-terpineol (Figure 13), etc. It is used to deal with acne lesions and extensively utilized in chemically unfastened mouth wash due to the fact that it fights against bad breath and dental plaque [7]. Figure 14 shows plants and plant oils that are used in cosmetic-containing herbal perfumes.

## 9. Bleaching Agents

### 9.1. Licorice Extract

It is largely the juice extracted from the root of the plant *Glycyrrhiza glabra* [37,38]. This compound is known to soak upthe UV-A rays and, in turn, acts as a strongpore and skin whitening agent. Glabridin [39] inhibited the enzyme tyrosinase [40], responsible for the manufacturing of melanin, which causes pore and skin pigmentation changes by nearly 50%. Licorice reverses the harm as a result of acne and has anti-inflammatory properties that provide a soothing effect on the pores and skin and minimizes redness [38].

### 9.2. Kojic Acid

It is obtained from a filamentous fungus called *Aspergillus oryzae* belonging to thecircle of relatives of Trichocomaceae. It inhibits the activity of tyrosinase [40], which synthesizes melanin. Moreover, it is used as a cancer prevention agent by specialists for remedying the harm occurred due to the sun or for treating delicate skin [41].

### 9.3. Phyllanthus Emblica

It has a significant quantity of antioxidant and antimicrobial characteristics. Moreover, gooseberry is abundant in vitamin C [42], which shields the pores and skin fromharm by the sun or proscribing hyperpigmentation. Gooseberry additionally has skin brightening properties that are well known or developed into existing pore and skin lighting synthetic compounds, for instance, hydroquinone. This prevents the pores and skin from oxidation stress, decrease progression of wrinkles in the skin, limits the formation of melanin, and empowers the skin to cling in the normal dampness [6,40,43].

### 9.4. GIGAWHITE

It is a licensed pore and skin lighting complex that is composed of seven Swiss snow-capped trees that may be naturally grown. These seven picked flora confirm the maximum elevated ability to inhibit the activity of the enzyme tyrosinase [40]. This prompts the production of GIGAWHITE to be applied as a common and more reliable replacement to hydroquinone for pores and skin in skin lightening merchandise [44].

### 9.5. Arbutin

It is obtained from the bearberry or Uvau. S. bush. Thise lement is known to have skin brightening properties. It treats sun burn damage and regulates melanogenesis [45].

### 9.6. Willow Bark Extract

It is beta hydroxy acid that facilitates to desquamate the skin and boost cell turnover. It consists of salicin, allowing new and healthy pores and skin cells to regenerate.

### 9.7. Allantoin

It is obtained from the flowering plants viz. *Symphytum officinale* or *S. uplandicum*. It is very adequate in the remedy of pores and skin burns, wounds, skin abrasions, etc.

## 10. Natural Colorants

### 10.1. Red Poppy Petals

It accommodates the dried whole or divided petals of *Papaver rhoeas* and are from the family Papaveraceae. Anthocyanins, including the gentiobioside of cyanidin, are responsible for the color [46].

### 10.2. Annatto

It is the reddish-orange colorant received from the seeds of the achiote tree, viz. *Bixa orellana*, family Bixaceae. Yellowish-orange shading is created via chemical substancessuch as bixin and norbixin. It is found in cosmetics and private care merchandis [46].

### 10.3. Red Beetroot

Pulverized pink beet root, *Beta vulgaris* (chenopodiaceae) is anextremely goodcolorant for primarily aqueous-based cosmetics. Betanin is the primary constituent present. Infusion of beetroot powder into glycerine creates a bright pink color, and it can be used inemulsions or water-based gels [46].

### 10.4. Blue Tansy

It is acquired from *Tanacetum annuum* plants from the family Asteraceae. The important chemical pigment is chamazulene (17–38%). To obtain a blue tingeinside the formulation, blue tansy oil can be used [46].

### 10.5. Butterfly Pea

Butterfly pea is also known as *Clitoria ternatea*. As the primary substance color, delphinidin, butterfly pea vegetation has great blue shading and much of the time it is utilized in herbal skincare [46].

### 10.6. Flavoxanthin

It is the to pcarotenoid extracted from the plant *Calendula officinalis*. Carotenoids are responsible for petal colorations in the yellow to purple range [47].

### 10.7. Beta-Carotene Pigment

It is isolated from the roots of *Daucus carota*. It is an orange colored pigment used in herbal skin care and takes on a delightful orange tinge [48,49].

### 10.8. Chamomile

Otherwise referred to as Hungarian or blue chamomile, this assortment of chamomile viz. *Matricaria recutita* yields fundamental oil that is dark is blue due to the fact of its excessive chamazulene content [32].

### 10.9. Saffron

It consists of the dried stigmas and tops of the forms of the *Crocus sativus* circle of relatives of Iridaceae. It consists of a number of carotenoids pigments, i.e., crocin and safranal [46].

### 10.10. Curcumin

It is the pigment removed from the dried rhizomes of *Curcuma longa* from the familyzingiberaceae. Curcumin offers a range of shades from yellow to profound orange. It is utilized in normal natural skin care be autifiers [50].

### 10.11. Carthamin

It is a yellow-colored pigment from the seeds of *Carthamus tinctorius*. It is used in various pores and skincare cosmetics.

### 10.12. Red Sandalwood

Biologically, it is referred to as *Pterocarpus santalinus* and consists of chemicals such as Santalin A or Santalin B, which are responsible for its crimson coloration, or Santalin Y, which is responsible for the yellow [46].

Few natural colorants are discussed in Table 5.

## 11. Antioxidants

### 11.1. Beta-Carotene

It is an herbal reddish–orange color in plants and fruits such as carrots and tomatoes. Beta-carotene [38] is a precursor (inactive from) to vitamin A. Beta-Carotene also canbe extracted from the algae *Dunaliella salina*. Beta-carotene is normally applied topores and skincare formulations for its capability to guard against UV-Alight-induced harm. Products infused with beta-carotene can assist in reducing the degrees of oxidative strain and contribute to ordinary enhancement of the skin’s appearance [48,56].

### 11.2. Vitamin C

Otherwise referred to as ascorbic acid, it is a nutrient found in distinctive nourishments and sold as a nutritional enhancement. Food containing vitamin C consists of citrus culmination, kiwifruit, broccoli, Brussel sprouts, uncooked bell peppers, and strawberries. Besides being an antioxidant, vitamin Chas different kin aids such as enhancing collagen formation and blurring dark spots [42,45,57,58].

### 11.3. Retinol (Diet A)

Retinol is especially successful because of its small shape, allowing it to infiltrate profoundly enough into the skin that it can viably sell collagen formation and quicken cell recovery, fixing and smoothing differences and wrinkles while enhancing pores and skin tone all at the same time. Retinol is a ground breaking cancer preventative agent for battling herbal aggressors that cause untimely aging of the skin [45,49].

### 11.4. Vitamin E

It is notably acquired for its excellent potential abilities among numerous organs within the body along with the skin. Aside from being ground breaking cell reinforcement, vitamin E is normally perceived for its ability to quicken the pores and skin’s restorative procedures. It is seldom found in lotions, creams, and salves, but it is figured to treat dry skins imply in it easement to decrease stretch marks [45].

### 11.5. Ginkgo

Biological Source: The leaves of ginkgo are acquired from the dioecious tree *Ginkgo biloba* from the family Ginkgoaceae. Its chemical constituents include diterpene lactones and flavonoids, kaempferol, quercetin, etc. [59]. It is used as an antioxidant that smoothens, rejuvenates, and promotes a youthful appearance [59].

### 11.6. Green Tea Extract

Both the intake and topical utility of ingredients rich in polyphenols aids in improving the skin’s natural resistance towards oxidative strain, prevents skin from aging, and may prevent skin malignancies [60].

### 11.7. Flavonoids

Green and black teas consist of antioxidants that can likely deal with rosacea and lessen infections and oxidative pressure. Flavonoids maintain mild UV and adjust signaling pathways that effect cellular functions to prevent aging a damage. It also assists in lowering collagen degradation, postponing aging, and inhibiting skin cancer [61].

### 11.8. Turmeric Oil

Biological Source: It is the oil acquired from dried rhizomes of the *Curcuma longa* from the family Zingiberaceae. Its chemical constituents include tetrahydrocurcuminoids. It is used as an antioxidant and in preventing aging [50].

### 11.9. Glutathione

It is a superb cancers prevention agent that allows for cell restoration and the safety of critical organs. It possesses an anti-melanogenic as set that causes pore and skin whitening. This “mother of all antioxidants” allows for the detoxification of the pores and skin and reduces the presence of wrinkles [61].

### 11.10. Coenzyme Q10

Increased topical use of this antioxidant facilitates combat against free radical damage and preserves the skin’s healthy cells. This nutrient is easily absorbed through the pores and skin and helps stimulate collagen manufacturing, which allows for improving the elasticity and texture [62,63].

### 11.11. Resveratrol

A chemical compound found mostly inside the skin acumination such as grapes and berries, peanuts, tea, and purple wine. This antioxidant boosts the activity of mitochondria, the powerhouse of the cell, promoting longer cell life in the body [64].

## 12. Protective Agents

### 12.1. Aloe

Biological Source: Aloe is dried juice obtained using incision from the base of diverse species of Aloe, viz. *Aloe vera* [62] or *Aloe barbadensis* from the family Liliaceae.

The main ingredients are aloin, barbaloin, β-barbaloin, and is obarbaloin. Other elements are aloe-emodin, emodin, resin, etc. [65]. It helps defend the skin against sunburn andharm from UV-A and UV-B rays. It is full of antioxidants and allowsfor healing sunburns much faster [65].

### 12.2. Jojoba Oil

Biological Source: It is a liquid waxy cloth articulated from an indigenous American shrub, *Simmondsia chinensis* [66]. It contains fatty acids such as ferulic acid, docosenoic acid, eleven-eicosenoic acid, oleic acid, palmitoleic acid, and erucic acid.

It has skin benefits such moisturizing properties. It is used to treat skin disorders, reduce wrinkles and signs and symptoms of aging. It reduces pore and skin inflammation, and works a san powerful cleanser. It also reduces darkcircles beneath the eyes, and it removes suntans. It is effective in treating acne [66].

### 12.3. Green Tea Extract

The tea plant *Camellia sinensis* includes four large polyphenolic catechins that include epicatechin, EGC, and EC-three-gallate. It was discovered that tea extract inhibited to a degree chemical carcinogenesis and carcinogenesis precipitated through UV-B [67].

### 12.4. Calendula

*Calendula officinalis* possesses outstanding antioxidant, anti-inflammatory, and wound recuperation properties. The essential oil of calendula consists of α-thujene, α-pinene, l,8-cineole, dihydro tagetone, and T-muurolol [68].

### 12.5. Carrot Seed Oil

Biological Source: This is extracted by acting as a distillation system on the dried seeds of *Daucus carota* [49], belonging to the circle of relatives of the family Apiaceae. Its chemical constituent includesalpha-pinene, beta-pinene, sabine, camphene, beta-bisabolene, geranyl acetate, and carotol [42,69]. As adefensive agent carrot seed oils not only enhance the pores and health of a healthy skin, but also found effective to repairs kin damage caused by pollutants and environmental strain. These oils were also reported with cleaning skin and making it more clear, brighter, and calmlytoned [42,49]. It also contributes to moisturizing [69].

### 12.6. Wheat Germ Oil

It is extracted from the germ of the wheat kernel, and it is specifically high in octacosanol. Wheat germ oil is exceptionally high in vitamin E. It also consists of fatty acids such as linoleic acid, palmitic acid, oleic acid, and linolenic acid. It helps to moisturize tissues and acts as an antioxidant to prevent free radical damage. It has a natural SPF score of 20 [42].

### 12.7. Witch Hazel

Biological Source: It comes from the dried eaves of *Hamamelis virginiana* from the circle of relatives of the family Hamamelidaceae.

Its chemical constituent include leaves, volatileoil (acetaldehyde and ionones), and flavonol glycosides (astragalin and myricitrin). Leaves additionally contain poly phenol components: procyanidin. The drug is utilized in cosmetology as Hamamelis water or distilled witch hazel extract in astringent education. It has antiseptic, anti-inflammatory, antibacterial and antifungal properties. It was reduces skin redness and offers remedy from irritated touchy skin [70].

### 12.8. Arnica

Empowered by way of homoeopathic experts for a longtime, *Arnica montana* has currently become a topical remedy in gels or creams formed to enhance inflammatory pores and skin conditions, reducing bruising, and healing continual wounds. The anti-inflammatory pastime of the dried flower heads of this plant is due to the presence of sesquiterpene lactones. Furthermore, it has been assessed as a treatment for hair loss because of and rogenetic alopecia, strain, and psychological causes [71,72]. Figure 15 shows list of plants used as protective agents in cosmetic-containing herbals.

## 13. Marketed Formulations

Various marketed formulations of cosmetics containing herbals are available in the market as shown Table 6.

## 14. Hair Cosmetic Containing Herbals

### 14.1. Horsetail/Snake Grass/Puzzle Grass

It is a herbaceous perennial plant known as Equisetum arvense, belonging to the circle of relatives of Equisetaceae [72]. The plant possesses several minerals inclusive of silicon, potassium, calcium, manganese, magnesium, phosphorus; phytosterols; dietary fiber; vitamin A, E, and C; tannins; alkaloids; saponins; flavonoids; glycosides; caffeic acid; phenolic ester.

It was reported as beneficial in hair development, treating dandruff and split-ends along with characteristic hair conditioner [78].

### 14.2. Coconut Oil

Coconut oil is obtained from the dried dense elements of the endosperm of coconut, *Cocos nucifera,* belonging to the family Palmae.

The oil carries approximately 95% saturated fatty acid with 8–10 carbon atoms. This indicates the presence of caprylic acid, (2%) capric acid, (50–80%), lauric acid, (27%), and myristic acid (1%).

It is used to prevent any kind of damage or damage to the hair, protection against dandruff, conditions the hair, acts as a tonic for hair, and additionally facilitates in styling of hair [78].

### 14.3. Amla

It consists of the fresh or dried fruit of *Emblica officinalis* belonging to the family Euphorbiaceae. It is rich in vitamin C, minerals, and amino acids. It is a staple in hair care products. It allows for color retention, makes the roots stronger, and promotes hairgrowth. It is a powerful treatment for hair loss and baldness [79].

### 14.4. Shikakai

Shikkai includes the fresh and dried fruits of *Acacia concinna* belonging to the family Leguminosae. Its chemical constituents are composed of saponins, flavonoids, tannins, and anthraquinone glycosides. It is traditionally utilized in Ayurvedic shampoos [80].

### 14.5. Reetha

Reetha is found in fresh or dried fruits (*Sapindus mukorossi*) belonging to the circle of relatives of Sapindaceae. The fruit consists of saponin mucilage. Sapindus saponin is amixture of six sapindosides (A, B, C, D, and mukorossi saponin (E1, Y1) such as diosin, proto diosgenin, gitoxigenin, chlorogenin, and ruscogenin. It has anti-inflammatory antimicrobial, and insecticidal interest. In cosmetics, it is used as a hair tonic [81].

### 14.6. Henna

Henna comes from the plant *Lawsonia inermis* belonging tothe family Lythraceae, and it incorporates a dye molecule known as lawsone, which produces henna powder when being processed. Other than lawsone, its distinctive constituents are gallic acid, glucose, mannitol, fats, resins (2%), mucilage, and a few alkaloids. Besides appearing as a hair coloring agent, it promotes hairgrowth, prevents dandruff and, additionally, conditions the hair. It also prevents itchy scalp [82].

### 14.7. Neem

Neem includes dried stem bark, root bark, leaves, and the end result, *Azadirachta indica*, belongs to the family Meliaceae [56]. Its chemical constituents contain tetranor terpenoid lactones azadirachtin, nimbin, nimbidin, salanin, and nimbolin-B. Neem oil additionally incorporates nimbolides, oli chinolide-B, and azadiradione [56,83]. It is commonly used for treatment of dandruff, because it produces antifungal, antibacterial, and ache-relieving compounds that treated and ruff [56].

### 14.8. Sandalwood Oil

Sandalwood oil is received via distillation of sandalwood, *Santalum album,* belonging to the family Santalaceae. The main odorous and medicinal constituent of sandalwood is santalol. It is present as a combination of alpha-santalol and betasantalol. It is used to stimulate hair growth and eliminated and ruff, and it additionally provides fragrance to the hair [84].

### 14.9. Brahmi

It is the fresh or dried herb of *Centella asiatica* belonging to the circle of relatives’ of Umbelliferae. It contains triterpenoid saponin glycosides, indocentelloside, brahmiaspect, brahminside, asiaticosides, Thankuni aspect, and isothankuniside.

It is used as are mark able hair growth promoter. Its regenerative properties as sistin the repair of hair follicles in addition to strengthening scalp tissue to encourage healthy hairgrowth. It also nourishes hair follicles and reduces hair loss [79].

### 14.10. Castor Oil

Castor oil is the constant oil received by the blood less expression of the seeds of *Ricinus Communis* belonging to the family Euphorbiaceae. It consists of glyceride of A ricinoleic, iso-ricinoleic, stearic, and dihydroxystearic acids [25]. It is used as a component in hair oils, as it moisturizes the scalp, eases dandruff and, additionally, complements thefitness of hair follicles and, inturn, promotes hairgrowth [25].

### 14.11. Babchi

*Psoralea corylifolia* (Babchi) is a critical plant with in the Indian Ayurveda and Tamil Siddha systemsof drugsand, additionally, Chinese medicine. The seeds of this plant incorporate a variety of coumarins including psoralen. *Psoralea corylifolia* extract contains some flavonoids, coumarins, and meroterpenes as well as excessive concentrations of genistein. It is used to prevent hair loss [85].

### 14.12. Carrot Seed Oil

It is used to deeply aid the scalp and hair to repairs plitends and contribute a soothing remedy for dehydration and inflammation together with itchiness resulting from harmful microorganisms, fungi, and dandruff. It also restores hair health and luster [69].

### 14.13. Witch Hazel Extract

It treats scalp sensitivity and provides comforts from signs and symptoms of itching and tenderness. It also reduces scalp irritation. It is a natural remedy for relieving sign sofa the scalp problems including dandruff and dryness [70].

### 14.14. Almond Oil

Almond oil is great for treating dandruff and hair damage. It additionally aids in scalp contamination and infection, and it is enriched with vitamins E and A, which might be helpful for both skin and hairhealth [22].

### 14.15. Lemongrass Oil

It is received from *Cymbopogon flexuosus* belonging to the family Poaceae, and it consists of noless than 75% aldehydes calculated as citral. It is the main source of citral from which ionone is derived. The oil also incorporates geraniol, linalool, limonene, alpha- and beta-pinene, myrcene, beta-phellandrene, etc. Having antimicrobial and antiseptic properties, it is used to deal with scalp irritation. Lemongrass oil strengthens hair follicles. It is known to fight hair loss while blended with rosemary and lavender, making it a top-not chair remedy [86]. All plant-based additives that are used in cosmetics are represented in Figure 16.

## 15. Oral Hygiene Product

### 15.1. Herbal Toothpaste

#### 15.1.1. Triphala/Jaiphal/Nutmeg

It is the kernel of the ripe dried seeds, *Myristica fragrans*, belonging to the Johncircle of relativesof Myristicaceae. The chemical composition of the oil includes clemicine, myristicin, geraniol, borneol, pinene, camphene, dipentene, eugenol and isoeugenol.

It is used in Ayurvedic toothpastes, as its essential oil has antibacterial properties, it helps in disposing of bacteria from the mouth, which may be responsible for causing badbreath [39].

#### 15.1.2. Clove Oil

It is a vital oil obtained via steam distillation of dried flower buds of *Eugenia caryophyllus*, belonging to the family Myrtaceae. It is 60–90% eugenol, which is the reason forits anesthetic and antiseptic properties. Its powerful antibacterial property relieves toothaches, and its spicyaroma imparts clean breath. It is prescribed to treat bleeding gums, toothaches, and dental caries [87].

#### 15.1.3. Neem Oil

Neem consists of the dried stem bark, root bark, leaves, and the end result, *Azadirachta indica* [7], belongs to the family Meliaceae. It contains tetranor, terpenoid, lactones azadirachtin, nimbin, nimbidin, salanin, and nimbolin-B. Neem oil additionally includesnimbolides, olichinolide-B, and azadiradione [56,83]. Its antibacterial of neem prevent gingivitis, and it is also used to prevent cavities and prevent bad breath [56].

#### 15.1.4. Miswak Extract

It is obtained from a tooth cleansing twigcrafted from the *Salvadora persica* tree belonging to the family Salvadoraceae.

Its main ingredients are beta-sitosterol, m-anisic acid, chloride salt of salvadora and gypsum, organic compounds together with pyrrolidine, pyrrole, and piperidine derivatives. It also contains flavonoids including kaempferol and quercetin; glycosides including salvadoside and salvadoraside [88]. Its resin has huge quantities of salts containing chlorine. It also has high amounts of fluorides, silica, sulphur, and vitamin C [42,69]. It has antibacterial properties that spare enamel decay and dental plaque formations [21].

#### 15.1.5. Eucalyptus Oil

It is vital oil received by means of steam distillation of sparkling leaves of *Eucalyptus globulus* and other species such as *E. polybractea*, *E. viminalis*, and *E. smithii* belonging to the circle of relatives of Myrtaceae. It consists of unstable oil of which 70–80% is 1,8-cineole, also referred to as eucalyptol. It also incorporates p-cymene, alpha-pinene, and flavonoids [16] inclusive of eucalyptin, hyperoside, and rutin It is powerful in opposition to cavities, dental plaques, gingivitis, and otherdental infections because of its germicidal properties [89].

#### 15.1.6. Myrrh Extract

Myrrh is an oleo-gum resin received from the stem of *Commiphora molmol*, belonging to the family Burseraceae. It contains alpha, beta, and gamma commiphoric acids. The volatile oil is a mixture of cuminic aldehyde, eugenol, cresol, pinene, limonene, dipentene, etc. Due to the fact of its antimicrobial houses, myrrh has traditionally been used to deal with oral infections and inflammations [90].

#### 15.1.7. Tea Tree Oil

It is important oil received by distillation from the leaves and terminal branch lets of *Melaleuca alternifolia*. Its chemical constituents include alpha-pinene, beta-pinene, sabinene, myrcene, alpha-phellandrene, alpha-terpinene, limonene, 1,8-cineole, p-cymene, linalool, terpinen-four-ol, and alpha-terpineol, etc. It is used as a natural antibacterial and anti-inflammatory within the oral cavity [7].

#### 15.1.8. Piper Betel Extract

Piper betel (vine) belongs to the family Piperaceae. Its chemical constituents are composed of chavibetol, caryophyllene, chavibetol acetate, f-pinene, eugenol, limonene, 1,8-cineole, etc. It forestalls the advancement of plaque development and disposes of microscopic organisms that cause bad breath [21].

#### 15.1.9. Tulsi Oil

Tulsi oil is acquired by way of distillation of sparkling and dried leaves of *Ocimum sanctum* from thefamilyLabiatae.Itcontainsapproximately70% eugenol, eugenol methyl ether, and also contains considerable amounts of Vitamin C. It has antibacterial properties and eliminates bad breath and, additionally, has enamel cleaning activities [42,57].

#### 15.1.10. Peppermint Oil

Peppermint Oil essential oil is produced by way of the distillation of *Mentha piperita* (Labiatae).

Its chief components are methyl acetate, isovalerate, menthone, cineole, inactive pinene, limonene, and 60–70% menthol. It is very effective in combating bad breath [91].

#### 15.1.11. Babhul

Its coagulated exudations obtained from the stems and branches of *Acacia arabica* (Leguminosae). It consists by and large of Arabian, (a combination of calcium, magnesium, and potassium salts of Arabic acid). It helps prevent swelling and bleeding of the gums, retention of gums, and healthy teeth strong [92].

#### 15.1.12. Vajradanti

It is also known as porcupine flower, and it is a species of the plant *Barleria prionitis* belonging to the family A canthaceae [93,94]. It is composed of different energetic chemical entities such as alkaloids, flavonoids, glycosides, phenolics, and saponins [94]. It is used to alleviate toothaches and to remedy bleeding gums [93].

#### 15.1.13. Majuphal

It is a species of oak, *Quercus infectoria*, bearing galls belonging to the family Fagaceae. The main phytoconstituents present in the plant are tannins, syringic acid, beta-sitosterol, amentoflavone, hexamethyl ether, methyl oleate, etc. It is used in the remedy of toothaches and gingivitis [95]. Natural plants used in herbal toothpaste are given in Figure 17.

### 15.2. Herbal Mouthwash

#### 15.2.1. Eucalyptus Oil

It is acquired by steam distillation of clean leaves of *Eucalyptus globulus* as well as from different species such as *E. polybractea*, *E. viminalis* and *E. smithii* belonging to thefamily Myrtaceae.

It is a volatile oil that presents out 70–80% 1,8-cineole also called eucalyptol. It also consists of p-cymene, alpha-pinene, and flavonoids, which includes eucalyptin, hyperoside, and rutin. It is used to facilitate the killing of bacteria that contribute to plaque, gingivitis, and bad breath [89].

#### 15.2.2. Clove Oil

It is acquired by steam distillation of dried flower buds of *Eugenia caryophyllus* from the family Myrtaceae. It has 60–90% eugenol, which is the reason for its anesthetic and antiseptic properties. It is used for its antibacterial activity that relieves toothaches, and its highly spicedaroma imparts fresh breath [87].

#### 15.2.3. Neem Oil

Neem comprises dried stem bark, root bark, and leaves and culminates in *Azadirachta indica* belonging to the family Meliaceae [56]. It contains tetranor, terpenoid, lactones, azadirachtin, nimbin, nimbidin, salanin, and nimbolin-B. Neem oil additionallyconsists of nimbolides, olichinolide-B, and azadiradione [83]. It has antibacterial properties that prevent enamel decay and dental plaque formation [56].

#### 15.2.4. Tea Tree Oil

It is an essential oil received through distillation of the leaves and terminal branchlets of *Melaleuca alternifolia*. Its chemical constituents include alpha-pinene, beta-pinene, sabinene, myrcene, alpha-phellandrene, alpha-terpinene, limonene, 1,8-cineole, p-cymene, linalool, terpinen-four-ol, and alpha-terpineol, etc. It is used for its natural antibacterial and anti-inflammatory effects in the oral cavity [96].

#### 15.2.5. Tulsi Oil

It is obtained from fresh as well as dried leaves of *Ocimum sanctum* belonging to the circle of relatives of Labiatae. It is composed of approximately 70% eugenol, eugenol methyl ether and, additionally, appreciable quantities of vitamin C [58]. It is antibacterial, eliminates bad breath, and also has to the cleaning properties [96].

#### 15.2.6. Peppermint Oil

It is acquired by distillation of *Mentha piperita*, from the circle of relatives of Labiatae. Its chief materials are methyl acetate, isovalerate, menthone, cineole, inactive pinene, limonene, and 60–70%menthol. It is a powerful agent in combating bad breath [96].

#### 15.2.7. Wintergreen Oil

It is acquired by distillation of the dried leaves of *Gaultheria procumbens*, from the family Ericaceae, but now it is distilled from the bark of the *Betula lenta* family Betulaceae. Methyl salicylate is the chief constituent of the oils haped from gaultherin on hydrolysis using the enzyme gaultherase inside the presence of water. It is used as an antibacterial and kills germs that cause bad breath and the gum ailment gingivitis [96].

## 16. Conclusions

Most people choose to use natural beautifying items for healthcare purposes. Now currently, there is an extremely large interest in everyday natural beauty care merchandise. Sound teeth, gleaming hair, and shining skin are noteworthy for the attractiveness of the human frame. Various herbal elements/additives used in natural formulations contain bleaching agents, fixed oils, perfuming agents, waxes, antioxidants, protective agents, herbal colorants, critical oils, and plant fabrics that include leaves, gums, mucilage, etc. Herbal formulations comprising herbal substances are side lining synthetic ingredients quite successfully. On the basis of data represented here indicated the use of herbal medicines and bioactive compounds for cosmetic purpose and treatment of various diseases and it relies upon on the correcting redient requirements and superiority of product. Herbal cosmetics must undergo proper control measures for protection as it is of outstanding significance.

## Figures and Tables

**Figure 1 molecules-27-00828-f001:**
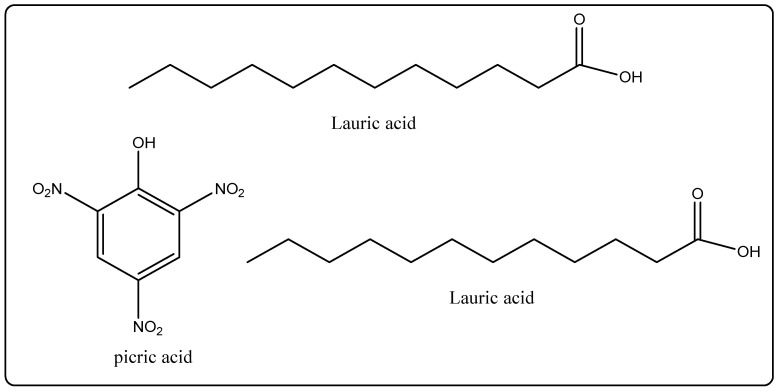
Pictorial representation of the chemical structures of chemical constituents of coconut oil.

**Figure 2 molecules-27-00828-f002:**
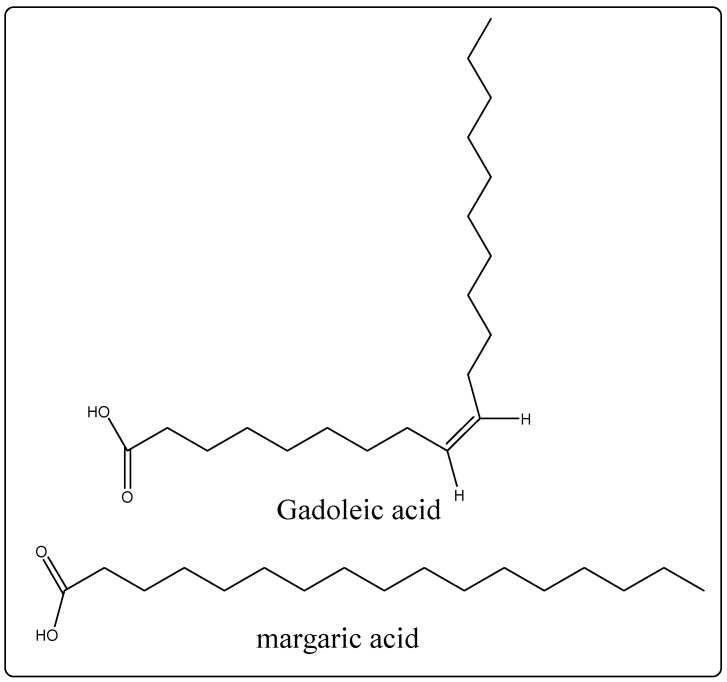
Chemical structure of chemical constituents present in the almond oil.

**Figure 3 molecules-27-00828-f003:**
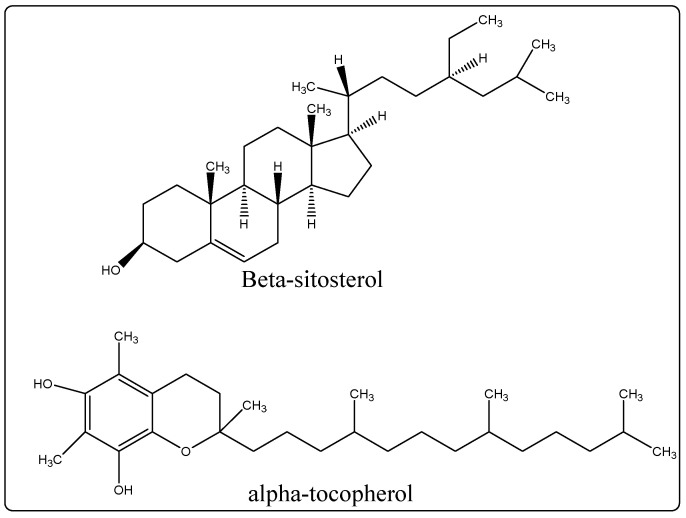
Chemical structure of chemical constituents present in the olive oil.

**Figure 4 molecules-27-00828-f004:**
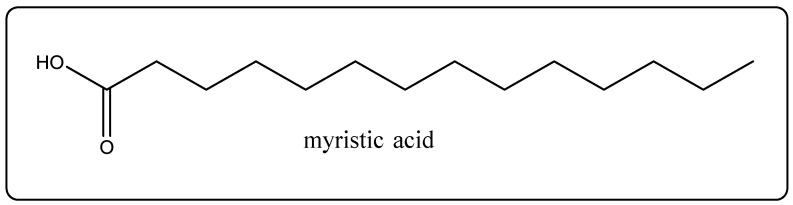
Chemical structure of myristic acid.

**Figure 5 molecules-27-00828-f005:**
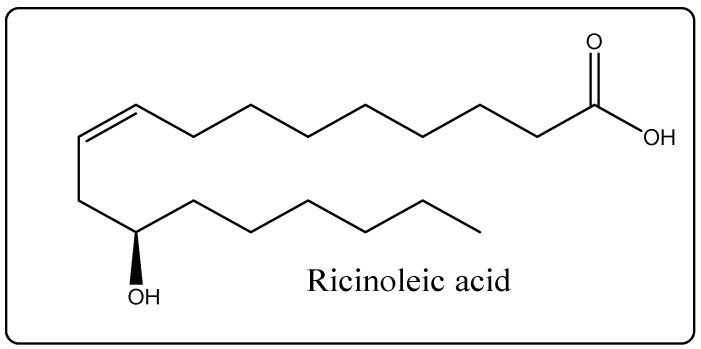
Chemical structure of ricinoleic acid.

**Figure 6 molecules-27-00828-f006:**
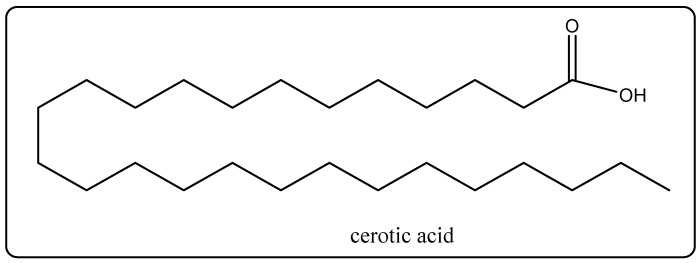
Chemical structure of cerotic acid.

**Figure 7 molecules-27-00828-f007:**
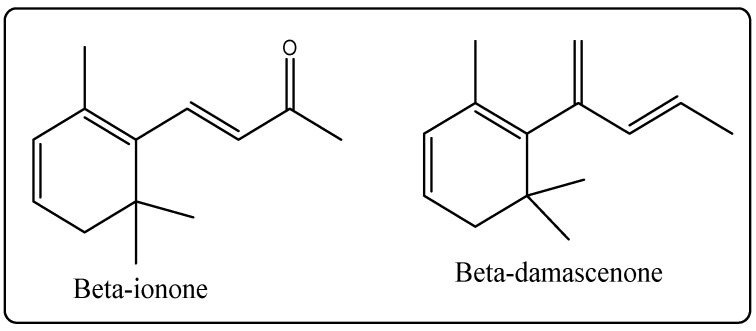
Chemical structure of the composition of rose oil.

**Figure 8 molecules-27-00828-f008:**
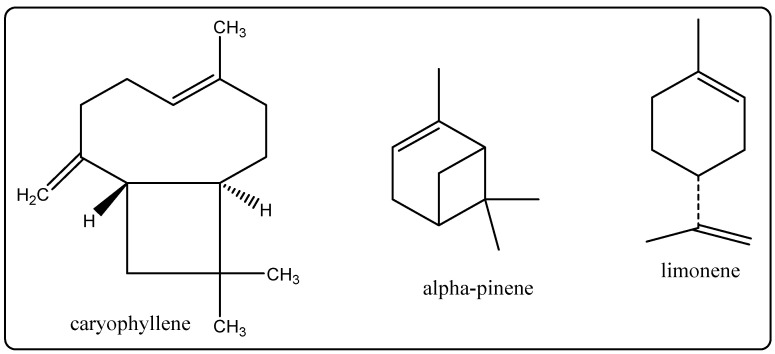
Chemical structure of the composition of lavender oil.

**Figure 9 molecules-27-00828-f009:**
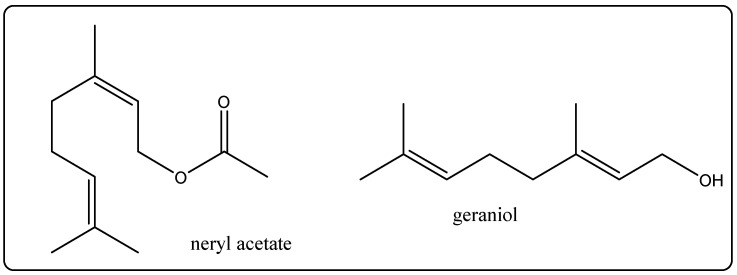
Chemical structure of the composition of Immortelle oil.

**Figure 10 molecules-27-00828-f010:**
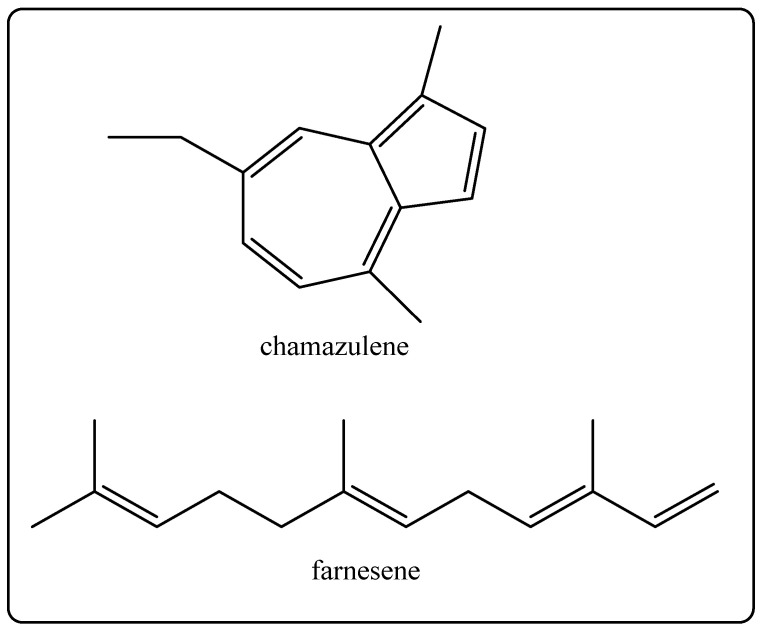
Chemical structure of the composition of chamomile oiloil.

**Figure 11 molecules-27-00828-f011:**
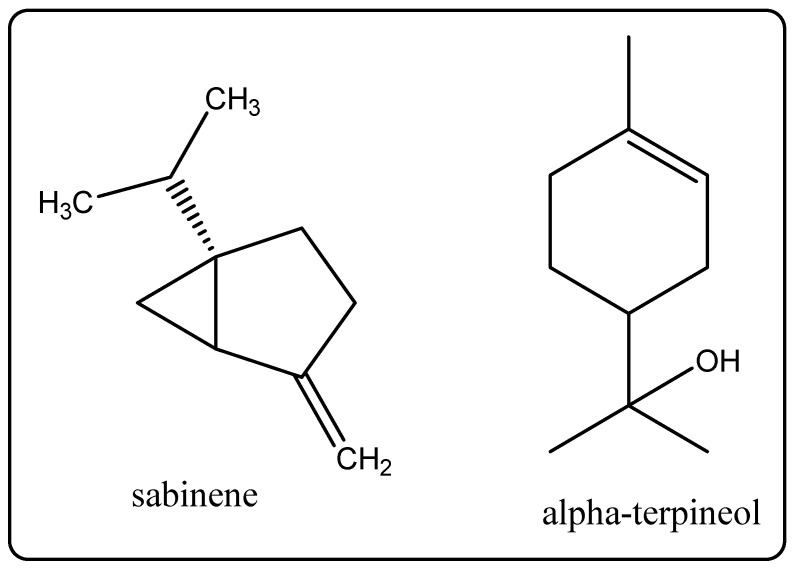
Chemical structure of the composition of neroli oil.

**Figure 12 molecules-27-00828-f012:**
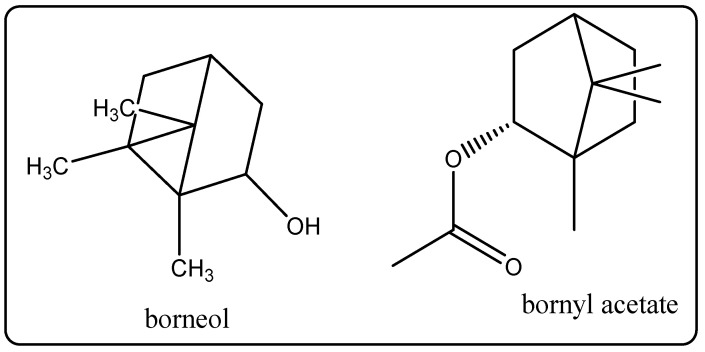
Chemical structure of the composition of rosemary essential oil.

**Figure 13 molecules-27-00828-f013:**
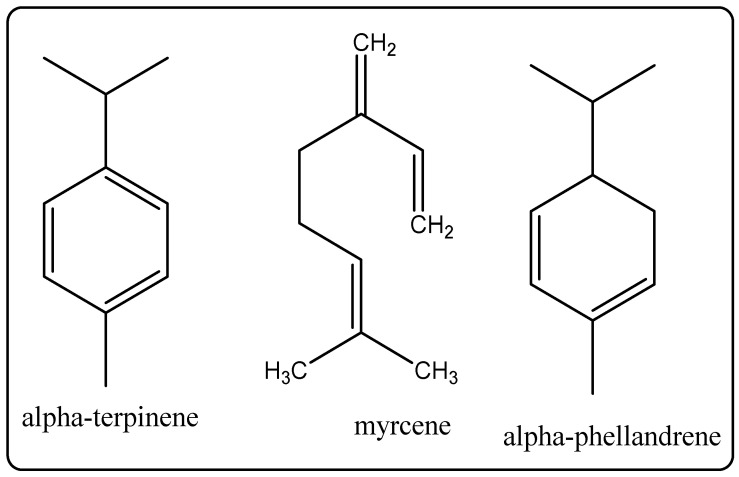
Chemical structure of the composition of tea tree oil.

**Figure 14 molecules-27-00828-f014:**
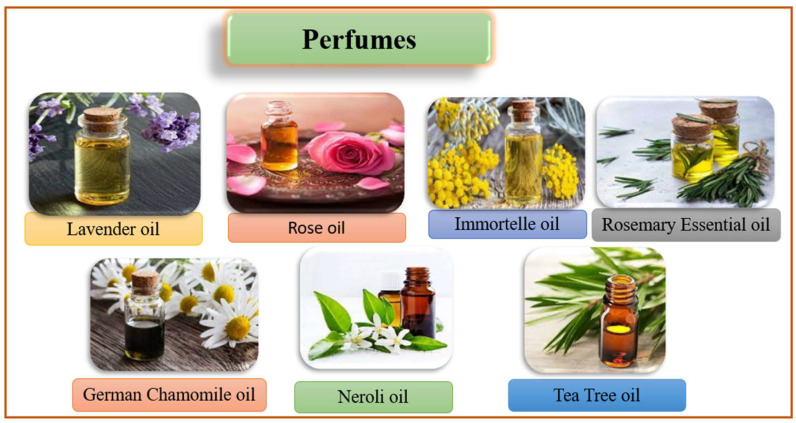
Illustration of plants oils used as perfume in cosmetic-containing herbals.

**Figure 15 molecules-27-00828-f015:**
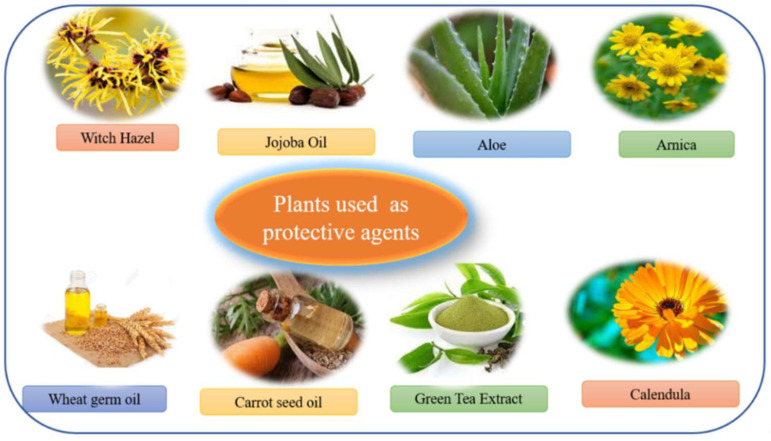
List of plants used as protective agents in cosmetic containing herbals.

**Figure 16 molecules-27-00828-f016:**
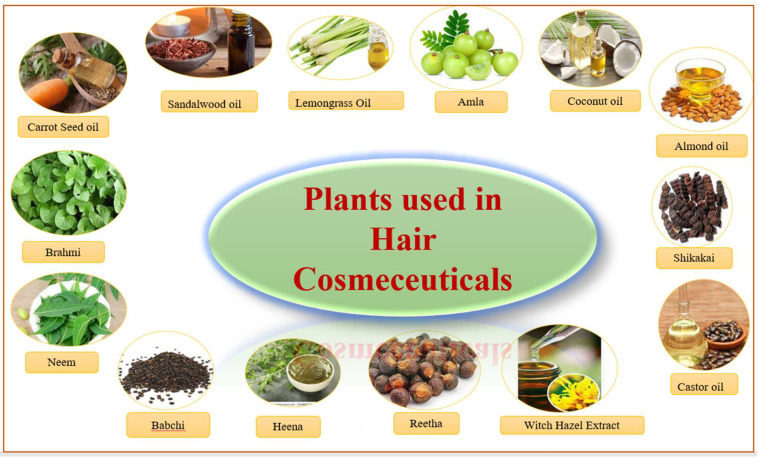
Plant-based additives used in hair cosmetic-containing herbals.

**Figure 17 molecules-27-00828-f017:**
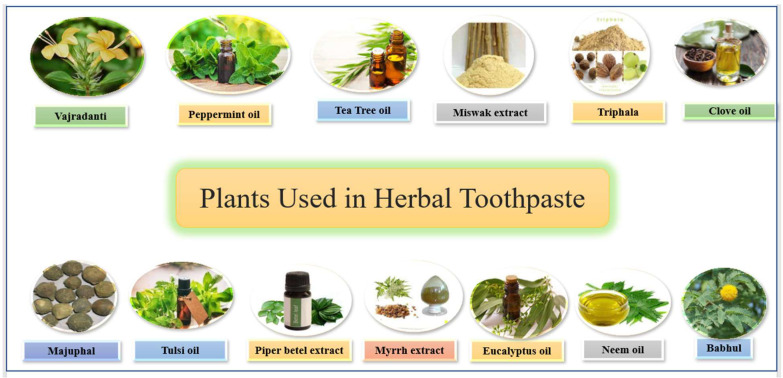
Plants and plant oils used in toothpaste.

**Table 1 molecules-27-00828-t001:** Various herbal additives and their role in cosmetics.

Name	Use in Cosmetics	References
Almond	Facial and body scrubs	[8]
Azadirachta	Toothpaste and skin care	[9]
Comfrey	Creams and lotions	[10]
Tulsi	Skin creams and lotions	[11]
Cucumber	Masks, toners, and cleansers	[12]
Henna	Dyeing of hair	[13]
Amla	Shampoo	[13]
Jasmine	Hair oil	[14]
Lemon	Skin tonics and cleansers	[14]
Apricot	Facial and body scrubs	[14]

**Table 2 molecules-27-00828-t002:** Types of skin and its care [1,17,18].

Type of Skin	Features of the Skin	Suitable Types of Skin Care		Reference
		Herbal	Essential Oils	
Normal skin	Even tone, texture is soft and smooth, no noticeable pores or flaws, and no oily fixes or flaky zones.	Juice of pomegranate leaves	Chamomile oil, Fennel oil, Geranium oil, Lavender oil, Lemon oil, Rose oil, and Sandalwood oil	[17]
Dry skin	Sebum level is lower and prone to sensitivity. Has a parched look andfeels tight. Chapping and cracking are signs of extremely dry, dehydrated skin.	Aloe vera, Calendula comfrey	Chamomile oil, Fennel oil, Geranium oil, Lavender oil, Lemon oil, Rose oil, Sandalwood oil, and Almond oil	[18]
Oily skin	Sparkling, thick, and dull shaded. Constantly sleek skin that has coarse pores and pimples and other imperfections. Inclined to clogged pores.	Aloe Vera, Burdock root, Chamomile, Horsetail, Oat straw, Thyme, Lavender, Lemon grass	Bergamotoil, Geraniumoil, Juniper oil, Lavender oil, Lemon oil, Sage oil, and Evening primrose oil	[17]
Combination	Dry or flaky, while the center part of the face, nose, chin and forehead (called the T zone) is oily.	Witch hazel, Menthol, Aloe vera, Turmeric, Wheat germ, Sweet flag	Citrus oils, Jasmine oil, and Sandalwood oil	[1]

**Table 3 molecules-27-00828-t003:** Types of skin problems and its herbal remedies [18,19].

Skin Problem	Features of the Skin	Remedies	References
Chapped skin	Rough texture and cracked skin.	St. John’s wort, olive oil, and mashed avocado after bathing or massaging with warm olive oil	[18]
Withered skin	Rough, wrinkles.	Carrot squeezed alongside a blend of egg white and honey	[19]
Sallow skin	No shading look, skin becomes dull and shows absence of essentialness. Responds rapidly to both warmth and cold.	Addition of Vitamin B to the diet	[19]
Sensitive skin	Burns from the sun and wind are common. Skin becomes dry and sensitive and is inclined to an unfavorably susceptible response.	Utilization of basic oils from chamomile, lavender, neroli, rose and sandalwood	[18]
Acne	Pockets of contamination that manifest as red sores, bubbles, and pimples.	Utilization of oil from red sandalwood	[19]

**Table 5 molecules-27-00828-t005:** Various natural colorants.

Source	Compound	Color	References
Indole derivatives			
*Murex brandaris* (Mollusk)	Bromoindigotin	Tyrian purple	[51]
*Indigo tinctorial* (Indigo)	Indigotin	Blue	[52]
Oxyindol glycoside			
*Beta vulgaris* (Beet root)	Betanin	Red	[53]
Diarylheptanoids			
*Curcuma longa* (Turmeric)	Curcumin	Yellow	[54]
Benzopyrones			
*Haematoxylon*(Logwood)	Haematin	Black	[54]
Carotenoids			
*Capsicum annum* (Capsicum)	Capsanthin	Orange–red	[55]
*Crocus sativus* (Saffron)	Crocin	Yellow–orange	[55]
*Tagetes erecta* (Marigold)	Lutein	Yellow	[46]
*Bixa orellana* (Annatto)	Bixin	Yellow–orange	[47]

**Table 6 molecules-27-00828-t006:** Marketed formulations containing herbal additives.

Ingredient	Action	Source	Marketed Preparation	References
Aloevera	Softens the skin	Aloe vera	Natures gel, Patanjali aloevera gel	[73]
α-Hydroxy acid	Exfoliates improves circulation And	Fruit acids (Glycolic acid, Lactic acid, Maleic acid, Citric acid)	Garnier anti-wrinkle cream, Olay anti-wrinkle cream, Aroma magic	[74]
Arnica	Astringent and soothing effect	*Arnica montana*	Arnica herbal cream	[75]
Β-Hydroxy acid	Antibacterial action	Salicylic (*Salix alba)* acid	Oxy med shampoo, skin Medica face cream	[74]
Boswellia	Anti-inflammatory and anti-aging	*Boswellia serrata*	Aroma Boswellia anti-wrinkle cream	[74]
Β-Carotene	Lessens peroxidation activity as an antioxidant	Carrot and tomato	Evion cream	[74]
Calendula	Soothes and softens skin, promotes cell formation	*Calendula officinalis*	Calendula cream	[74]
Centella	Skin conditioner, boosts collagen production, improves texture of the skin, reduces stretch mark	*Centella asiatica*	Keratin complex cream, Estee Lauder	[76]
Coriander seed oil	Anti-inflammatory action and skin-lightening properties	*Coriander sativa*	Richmond nature cream	[76]
Cucumber	Antioxidant, refreshes, and tighten pores	*Cucumis sativus*	Everyuth cream	[77]
Green tree extract	Antioxidant	Green tea	Alba botanica moisturizer	[76]
Kinetin	Free radical scavenger as well as antioxidant	Yeast	Kinerase pro therapy	[77]
Licorice extract	Bleaching agent/skin brightening	*Glycyrrhiza glabra*	Liquorice balm	[76]
Neem oil	Antimicrobial properties	*Azadirachta indica*	Himalaya neem face wash	[74]
Rosemary extract	Antioxidant and antimicrobial action	*Rosemarinus officinalis*	Loreal body cream	[76]
Turmeric	Antioxidant and antimicrobial action	*Curcuma longa*	Vicco turmeric cream	[74]
Vitamin A	Antioxidant	Vitamin A, C, E (lemon, citrusfruit, essential oils)	Everyuth peel	[75]
Witch hazel extract	Tones skin	*Hamamelis virginiana*	Thayers skintoner, Witch hazel cream	[76]

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
