# Peer review of "Bioactive-Based Cosmeceuticals: An Update on Emerging Trends"

_molecules, 2022, doi:10.3390/molecules27030828_

Round 1

Reviewer 1 Report

Bioactive Based Cosmeceuticals: An Update on Emerging Trends

This review needs some further modification.

The authors claimed that:

Cosmeceuticals, a cosmetic that has or is claimed to have medicinal properties.

Cosmeceuticals are cosmetic products with bioactive ingredients purported to

have medical benefits. There are no legal requirements to prove that these products

live up to their claims. The name is a combination of "cosmetics" and "pharmaceuticals".

Pls see my comments:

Based on the guidelines by FDA: https://www.fda.gov/cosmetics/cosmetics-labeling-claims/cosmeceutical

“Consumers and manufacturers sometimes have questions about the term "cosmeceutical."

The term "cosmeceutical" has no meaning under the law. While the Federal Food, Drug, and Cosmetic Act (FD&C Act) does not recognize the term "cosmeceutical," the cosmetic industry uses this word to refer to cosmetic products that have medicinal or drug-like benefits.

A product can be a drug, a cosmetic or both. The FD&C Act defines drugs as those products that cure, treat, mitigate or prevent disease or that affect the structure or function of the human body, if a product makes such claims it will be regulated as a drug. Cosmetics are intended to beautify, promote attractiveness, alter appearance or cleanse; they are not approved by FDA for sale nor are they intended to effect structure or function of the body.”

My suggestion: Delineate the term cosmeceutical with cosmetic containing herbal  as the author has stated that it is “plants, their part used, active ingredient and the therapeutic properties associated with the same”

The statements are confusing:

  1. The present review highlighted the use of natural products in cosmeceuticals, as natural products have number of health benefits over their side effects to the mammals.
  2. Mainly, the utilization of phytoconstituents in the care of skin and hair, like dryness, acne, eczema, inflammation of skin, aging, hair growth, dandruff and hair colourant has been explained.
  3. Cosmeceuticals, the maximum rapidly growing department in the natural splendor industry
  4. These are topical corrective pharmaceutical combos proposed to complementthe beauty by way of the utilization of those components which have brought fitness associated characteristic or benefit. (fitness??)
  5. Gets easily merged with hair and the skin. (merged??)

Author Response

Reply to reviewer’s comments

Dear sir/mam

We are pleased to have your valuable suggestions for our review article entitled “Bioactive Based Cosmeceuticals: An Update on Emerging Trends”.

Kindly finds the answers to the suggestions given by the reviewers:

Reviewer 1

Comment 1: Delineate the term cosmeceutical with cosmetic containing herbal as the author has stated that it is “plants, their part used, active ingredient and the therapeutic properties associated with the same”.

Answer to comment: As per the suggestions given by the reviewer, the term cosmeceutical has been delineated with cosmetic containing herbal.

Comment 2: The statements are confusing:

The present review highlighted the use of natural products in cosmeceuticals, as natural products have number of health benefits over their side effects to the mammals.

Mainly, the utilization of phytoconstituents in the care of skin and hair, like dryness, acne, eczema, inflammation of skin, aging, hair growth, dandruff and hair colourant has been explained.

Cosmeceuticals, the maximum rapidly growing department in the natural splendor industry

These are topical corrective pharmaceutical combos proposed to complement the beauty by way of the utilization of those components which have brought fitness associated characteristic or benefit. (fitness??)

Gets easily merged with hair and the skin. (merged??)

Answer to comment: All the statements suggested by the reviewer has been corrected accordingly in order to clarify the information to the readers. The amendment made has been highlighted in yellow colour.

Waiting for hopeful consideration.

Thanking you, Regards

Reviewer 2 Report

i would like to appreciate to all authors for their efforts, in this review they tried to describe the bioactive compounds used in cosmetics for different purposes in different aspects. However I do have following observations.

  1. In generally, i observed many flaws in the entire manuscript. It require extensive modification, formatting and english revision. Needs consistency in writing and information, connectivity among sections, standard presentation.
  2. In all tables, please put individual reference for individual additive/ source/ compound etc.
  3. section 2 and 3, difficult to understand, please rewrite these with clear description.

Author Response

Reviewer 2

Comment 1: In generally, I observed many flaws in the entire manuscript. It require extensive modification, formatting and English revision. Needs consistency in writing and information, connectivity among sections, standard presentation.

Answer to comment: As per suggestion, the manuscript has been throrougly checked for connectivity, and grammatical errors.

Comment 2: In all tables, please put individual reference for individual additive/ source/ compound etc.

Answer to comment: Individual Reference has been added to their respective places in each table, as recommended by the reviewer.

Comment 3: section 2 and 3, difficult to understand, please rewrite these with clear description.

Answer to comment: As per the comments suggested by the reviewer, both these sections has been written again, in order to provide more clarity, highlited by yellow colour.

Kindly find the solutions for comments (suggested by reviewers) in the revised manuscript entitled “Bioactive Based Cosmeceuticals: An Update on Emerging Trends”.

Waiting for hopeful consideration.

Thanking you

Regards

Round 2

Reviewer 1 Report

All comments have been addressed. 

Author Response

Dear sir/mam

We are pleased to have your valuable suggestions for our review article entitled “Bioactive Based Cosmeceuticals: An Update on Emerging Trends”.

Kindly finds the answers to the suggestions:

  1. However, please make sure that the size and orientation of the chemical structures in the figures are strictly consistent.

Answer: All the chemical structures has been reviewed for size and orientation to maintain consistency.

  1. Short cover-letter

Answer: cover letter has been provided as attached files.

Awaiting for hopeful consideration.

Thanking you

Regards